# An Unusual Two-Domain Thyropin from Tick Saliva: NMR Solution Structure and Highly Selective Inhibition of Cysteine Cathepsins Modulated by Glycosaminoglycans

**DOI:** 10.3390/ijms25042240

**Published:** 2024-02-13

**Authors:** Zuzana Matoušková, Katarína Orsághová, Pavel Srb, Jana Pytelková, Zdeněk Kukačka, Michal Buša, Ondřej Hajdušek, Radek Šíma, Milan Fábry, Petr Novák, Martin Horn, Petr Kopáček, Michael Mareš

**Affiliations:** 1Institute of Organic Chemistry and Biochemistry, Czech Academy of Sciences, Flemingovo n. 2, 16610 Praha, Czech Republic; zuzana.matouskova@uochb.cas.cz (Z.M.); katarina.orsaghova@uochb.cas.cz (K.O.); michal.busa@uochb.cas.cz (M.B.); milan.fabry@uochb.cas.cz (M.F.); martin.horn@uochb.cas.cz (M.H.); 2Department of Biochemistry, Faculty of Science, Charles University, Hlavova 8, 12800 Praha, Czech Republic; 3First Faculty of Medicine, Charles University, Katerinska 32, 12108 Praha, Czech Republic; 4Institute of Microbiology, Czech Academy of Sciences, Prumyslova 595, 25250 Vestec, Czech Republic; 5Institute of Parasitology, Biology Centre, Czech Academy of Sciences, Branisovska 31, 37005 Ceske Budejovice, Czech Republic; 6Biopticka Laborator, Mikulasske Namesti 4, 32600 Plzen, Czech Republic

**Keywords:** cathepsin, cysteine protease, tick, parasite, saliva, thyropin, protease inhibitor, protein structure

## Abstract

The structure and biochemical properties of protease inhibitors from the thyropin family are poorly understood in parasites and pathogens. Here, we introduce a novel family member, Ir-thyropin (IrThy), which is secreted in the saliva of *Ixodes ricinus* ticks, vectors of Lyme borreliosis and tick-borne encephalitis. The IrThy molecule consists of two consecutive thyroglobulin type-1 (Tg1) domains with an unusual disulfide pattern. Recombinant IrThy was found to inhibit human host-derived cathepsin proteases with a high specificity for cathepsins V, K, and L among a wide range of screened cathepsins exhibiting diverse endo- and exopeptidase activities. Both Tg1 domains displayed inhibitory activities, but with distinct specificity profiles. We determined the spatial structure of one of the Tg1 domains by solution NMR spectroscopy and described its reactive center to elucidate the unique inhibitory specificity. Furthermore, we found that the inhibitory potency of IrThy was modulated in a complex manner by various glycosaminoglycans from host tissues. IrThy was additionally regulated by pH and proteolytic degradation. This study provides a comprehensive structure–function characterization of IrThy—the first investigated thyropin of parasite origin—and suggests its potential role in host–parasite interactions at the tick bite site.

## 1. Introduction

The thyroglobulin type-1 (Tg1) domain is typically a 7 kDa protein module found in a variety of unrelated multidomain proteins in the animal kingdom. It was first identified in thyroglobulin, which carries eleven Tg1 repeats [1]. The Tg1 domains, which feature a conserved pattern of cysteine residues, are divided into subgroups 1A and 1B based on the number of cysteines [1]. The variable loops in the molecular architecture of the Tg1 domain make it highly adaptive and account for the multiple functions it has acquired through evolution, including protease inhibition. Tg1 domain-containing proteins with antiproteolytic activity are known as thyropins [2], classified as protease inhibitor family I31 by the Merops database [3]. While Tg1 domains have been shown to directly inhibit cysteine and aspartic proteases, mainly those of the cathepsin type, individual thyropins differ greatly in their inhibitory specificities [4].

Most of our knowledge about thyropins and their inhibitory activity derives from the following members: the p41 invariant chain fragment, involved in major histocompatibility complex (MHC) class II maturation and antigen processing [5,6]; saxiphilin, a neurotoxin-binding protein from bullfrogs [7,8]; equistatin, derived from sea anemones, which contains domains that selectively inhibit cysteine or aspartic cathepsins [9,10]; human testican-1, a multi-domain proteoglycan from the brain [11,12]; human epithelial cell adhesion molecule (EpCAM), a carcinoma cell marker [13]; and the cysteine protease inhibitor ECI from chum salmon eggs [14]. Additionally, Tg1 domains in insulin-like growth factor binding proteins (IGFBPs) have been proposed as proteolysis regulators [15,16]. 

Of the thyropins whose inhibitory properties have been demonstrated, only two have been characterized at the structural level. These include crystal structures of saxiphilin, which has two Tg1 domains embedded in a large transferrin architecture [8], and the p41 fragment, a single Tg1 domain thyropin, in complex with cathepsin L [6]. This complex identified the reactive center on the Tg1 domain, which is formed by three loops that bind to the active site of cathepsin L. In addition to their inhibitory effects, Tg1 domains may establish more complex relationships with the target cathepsin. For instance, as demonstrated for cathepsin L complexed with the p41 fragment, the Tg1 domain may stabilize the protease at a neutral pH [17]. Another example is the proteolytic degradation of testican-1, an inhibitor of cathepsin L, by the same protease at high concentrations [12].

There is very little information on the structure–function relationships of thyropins in invertebrates, particularly those found in parasites. To address this gap, we performed a molecular characterization of thyropin from the hard tick *Ixodes ricinus*, a well-known vector of Lyme borreliosis and tick-borne encephalitis. Although thyropins have been recently identified by transcriptomic and proteomic analyses in several tick species [18,19,20,21], their biological roles remain unknown. Our interest in thyropins builds upon successful research into another family of protease inhibitors targeting cysteine cathepsins, the cystatins. These proteins found in tick saliva act as powerful immunomodulators of the host response, while those in the tick gut regulate blood digestion [22,23,24,25].

In this study, we provide a comprehensive functional and structural characterization of Ir-thyropin (IrThy), an unusual thyropin secreted in the saliva of *I. ricinus* ticks. Its small molecule consists of only two Tg1 domains, which are atypically disulfide-bonded. The narrow inhibitory specificity of IrThy was explained by NMR structural analysis and was further shown to be modulated by glycosaminoglycans (GAGs). Finally, we propose a role for IrThy in the initial stages of host–parasite interactions at the tick bite site.

## 2. Results

### 2.1. IrThy—A Salivary Protein Secreted by the I. ricinus Tick

Based on the analysis of the salivary gland transcriptome (sialome) of the hard tick *I. ricinus* [26], we identified a set of homologous sequences belonging to the I31 family of thyropin proteins. Using a consensus sequence, we designed primers for PCR amplification from the cDNA of salivary glands of half-fed *I. ricinus* females. The resulting amplicon was cloned and sequenced, yielding a sequence with an open reading frame of 501 bp (Appendix A). This sequence was deposited in GenBank under accession number PP107940. The encoded sequence of the protein, named IrThy, exhibited ~97% and ~96% identity to GenBank accessions JAA67758.1 and JAP74612.1, respectively, from *I. ricinus* [26,27].

To demonstrate the specific localization of IrThy, we applied an LC–MS/MS strategy based on enzymatic digestion of a complex proteome in conjunction with MS/MS peptide sequencing. Based on our analysis of salivary gland extract and saliva from *I. ricinus* females (Appendix A), we identified IrThy in both materials, indicating that IrThy was a secreted salivary protein.

Next, we compared the mRNA transcript levels of IrThy in two tissues that play a key role in the tick feeding process (Figure 1A). Specifically, we observed substantially higher expression in the salivary glands of half-fed females than in the midgut, which is responsible for blood digestion. Figure 1B shows the dynamics of IrThy expression during tick feeding on the host. Unfed females displayed the highest level of expression, which then gradually decreased to reach an expression level approximately one order of magnitude lower than in fed females. 

In summary, we cloned IrThy from *I. ricinus* as a new member of the thyropin family. It is predominantly expressed in the salivary glands of unfed female ticks and is secreted into the tick saliva, suggesting its involvement in the early stages of tick–host interaction. 

### 2.2. IrThy Is a Two-Domain Thyropin: Sequence Analysis, Evolutionary Distribution, and Recombinant Production

IrThy cDNA contains an open reading frame encoding a protein of 166 amino acid residues. It contains a 19-residue signal peptide followed by a mature IrThy sequence of 147 residues consisting of two consecutive Tg1 domains from the I31 thyropin family (Figure 2A). The N- and C-terminal domains have 79 and 68 residues, respectively, and share 26% sequence identity. Both domains belong to Tg1 subtype 1A [1], which features three disulfide bonds. By evaluating the connectivity of the disulfides by LC–MS/MS peptide mapping (Appendix A), we found that all of the cysteines formed disulfide bridges. The connectivity is as follows: Cys17–Cys37, Cys48–Cys59, and Cys61–Cys81 in the N-domain, and Cys79–Cys103, Cys115–Cys139, and Cys122–Cys147 in the C-domain (Figure 2A). Two of them are associated with the central sequence Cys79(N-domain)–Lys80–Cys81(C-domain), where Cys79 is cross-linked to the C-domain and Cys81 to the N-domain; therefore, they can be classified as interdomain disulfide bridges (Figure 2A). Sequence comparison shows that the N-domain of IrThy shares the disulfide pattern with the prototype thyropin, the human p41 fragment, while the pattern in the C-domain is modified (Appendix A).

Given that IrThy consists of two Tg1 domains, we decided to investigate the phylogenetic distribution of IrThy homologs with analogous domain compositions. The Pfam database (hosted by InterPro) [28] lists over 20,000 proteins containing the Tg1 domain (Pfam entry: PF00086). These proteins are typically multi-domain, with up to more than a dozen Tg1 repeats, often combined with domains of other types. Figure 2C shows the distribution of Tg1 domain-containing proteins for three different search parameters. Selection I presents all sequences that have two Tg1 domains along with other Pfam domains. Most of them, 77%, belong to vertebrates, and 23% are mammalian. Selection II differs from Selection I in that it excludes proteins containing other Pfam domains. This resulted in a predominance of invertebrate sequences (70%), with over half coming from arthropods (57%). In Selection III, we restricted the sequence length to 180 residues spanning two consecutive Tg1 domains; this is consistent with the domain pattern of IrThy. These short two-domain thyropins were mostly found in invertebrates (63%), including ticks; none were found in mammals. This indicated that IrThy-like thyropins from ticks have no analogs in mammalian hosts.

Recombinant full-length IrThy was prepared as a mature protein with a removable oligohistidine tag (ultimately cleaved by a TEV protease) and expressed using the S2 insect cell system. The same procedure was used to produce individual single domains of IrThy, namely the N-domain protein IrThy-Nd and the C-domain protein IrThy-Cd. For simplicity, interdomain disulfides linked to the central sequence Cys79–Lys80–Cys81 were replaced by intradomain disulfides spanning a sequence two amino acids shorter, namely Cys61–Cys79 in IrThy-Nd and Cys81–Cys103 in IrThy-Cd. These disulfides formed correctly in the expressed single-domain proteins, as confirmed by LC–MS/MS peptide mapping (Appendix A). All three recombinants were purified to homogeneity from expression media (see Section 4). They migrated on SDS–PAGE as single bands corresponding to the predicted molecular masses of 16.9, 9.6, and 7.9 kDa for IrThy, IrThy-Nd, and IrThy-Cd, respectively (Figure 2B). To increase the homogeneity of the recombinants for structural analysis, the N-glycosylation motif on the C-domain was mutated (Figure 2A); this motif is located on the opposite side of the domain from the predicted reactive center for protease binding (see Section 2.6.).

### 2.3. The Unique, Narrow Inhibitory Specificity of IrThy

Purified recombinant IrThy and its individual domains, IrThy-Nd and IrThy-Cd, were screened in vitro for inhibitory activity against two panels of proteases, including archetypal representatives of major protease classes and relevant cysteine cathepsins of the mammalian host. The latter panel, selected to cover a wide range of endo- and exopeptidase cleavage specificities of cysteine cathepsins, consisted of cathepsins F, K, L, S, and V (endopeptidases), cathepsins B and X (a carboxydipeptidase/endopeptidase and carboxypeptidase, respectively), and cathepsins C and H (an aminodipeptidase and aminopeptidase, respectively).

Inhibition constants (*K*_i_) were determined using a fluorescence activity assay; the specificity profiles are compared in Table 1. No inhibition was observed for members of aspartic and serine proteases or for cysteine proteases of the CD clan. Inhibition was observed only for certain cysteine cathepsins with endopeptidase activities from the CA clan of cysteine proteases. In general, IrThy and its two individual domains selectively inhibited cathepsins V, K, and L as follows: *K*_i_ values were within the range of 27–41 nM and 44–57 nM for cathepsin V and cathepsin K, respectively; cathepsin L was more sensitive to inhibition by IrThy and IrThy-Nd (*K*_i_ of 201 and 179 nM, respectively) than by IrThy-Cd (*K*_i_ of 795 nM). However, IrThy-Nd exhibited a broader inhibitory specificity and, in addition to the three cathepsins mentioned above, it effectively inhibited two other proteases with endopeptidase activities from the CA clan, namely cathepsin F and the archetypal plant protease papain, with *K*_i_ values of 207 and 153 nM, respectively.

We demonstrated that IrThy is capable of inhibiting the activity of model target proteases across the entire pH activity range and, moreover, that the degree of inhibition is pH-dependent (Figure 3A). The most effective inhibition was observed at a mildly acidic pH of ~6, with reduced inhibitory potency at more acidic and neutral/alkaline pH values. The regulatory effect of pH was more pronounced for cathepsin V than for cathepsin K. We also analyzed the inhibition mode of IrThy, classifying it as competitive, which suggests that IrThy binds in the protease active site as previously described for the p41 fragment [6]. The inhibition mode was determined using a Lineweaver–Burk plot with a series of straight lines intersecting on the 1/v axis (Figure 3B, C). 

In summary, we demonstrated that IrThy is a thyropin with a unique and narrow inhibitory specificity toward mammalian cysteine cathepsins. It targeted only the endopeptidase cathepsins V and K within nanomolar range, followed by cathepsin L; the other cathepsins with the spectrum of endopeptidase and exopeptidase activities proved insensitive to IrThy. 

### 2.4. IrThy Is Prone to Proteolytic Degradation by Target Proteases at High Concentrations

The Tg1 domains proved susceptible to cleavage by cysteine cathepsins, as reported in [29]. Human testican, a cathepsin L inhibitor from the thyropin family, undergoes degradation by its target protease, cathepsin L, at high concentrations [12]. In this context, we chose to examine the sensitivity of IrThy to the proteolytic activity of cathepsins V, K, and L, which are selectively inhibited by IrThy (see Section 2.3). For comparison, we also included an experiment with cathepsin B as a model endopeptidase/exopeptidase, which is not targeted by IrThy.

First, we applied the concentration conditions used in the kinetic assay for measuring the inhibitory potency of IrThy (Figure 4). No apparent degradation of IrThy was observed using these catalytic concentrations of cathepsins, with an enzyme:inhibitor ratio of 1:20,000 (*w*/*w*) (corresponding to 0.2 nM:6 µM concentrations). On the contrary, incubation of IrThy with cathepsin concentrations increased by three orders of magnitude (1:5 ratio, *w*/*w*) (corresponding to 0.8 μM:6 µM concentrations) led to the substantial degradation of IrThy by all tested cathepsins. The time course of the degradation reaction is shown in Appendix A, which illustrates the generation of IrThy fragments. Furthermore, we used mass spectrometry for mapping the cleavage sites, which were found to be widely distributed on the IrThy molecule (Appendix A). 

To conclude, we provide evidence that IrThy can be proteolytically degraded by both its target and non-target cysteine cathepsins. We speculate that this may represent a concentration-dependent regulatory mechanism that controls the physiological inhibitory effect of IrThy.

### 2.5. Complex Glycosaminoglycan Modulation of IrThy Inhibitory Activity

As previously reported, GAGs, which are widely distributed in tissues, modulate interactions between several protease inhibitors of proteinaceous character and their target proteases, including cathepsins [30,31]. However, this relationship has not been investigated in thyropins. To address this shortcoming, we tested the effect of GAGs on IrThy action against cathepsins V, K, and L, which proved sensitive to IrThy inhibition (Table 1). For the screening, we used a panel of GAGs that included (i) heparin (17–19 kDa) and its disaccharide fragment (0.7 kDa), (ii) chondroitin-4-sulfate, chondroitin-6-sulfate, dermatan sulfate (all in the 20–60 kDa range), and (iii) dextran sulfate (9–20 kDa), which served as a model sulfated polysaccharide GAG analog. Individual GAGs at a concentration of 10 µg/mL were incubated with IrThy and a target cathepsin. The resulting change in cathepsin activity was measured using a fluorescence activity assay and compared to the control experiment without GAG. 

The results in Figure 5A show the dramatic effects of GAGs on the inhibition of cathepsins K and V. The inhibitory potency of IrThy against cathepsin K was substantially reduced from ~50% inhibition to ~7, 16, and 23% in the presence of heparin, chondroitin-4-sulfate, and chondroitin-6-sulfate, respectively, and was completely abolished by dermatan and dextran sulfates. The opposite effect was observed for cathepsin V, whose inhibitory potency was greatly enhanced from ~25% to within an inhibitory range of 55–70% in the presence of all tested GAGs. The only exception was the heparin disaccharide fragment, which had no significant effect on IrThy inhibition of cathepsin V or the other cathepsins. In contrast to cathepsins K and V, the inhibition of cathepsin L by IrThy was not substantially affected by GAGs in general; changes in inhibitory potency were within 15% (Figure 5A).

Figure 5B shows the dependence of the modulatory effect on the GAG concentration as demonstrated by heparin. For the inhibition of cathepsins K and V by IrThy, heparin was highly effective at concentrations of 1 µg/mL and above, where the inhibitory potency of IrThy reached its minimum value against cathepsin K and its maximum value against cathepsin V. Heparin concentrations at and below 0.1 µg/mL had a much less pronounced effect (<17% compared to the control without heparin). Inhibition of cathepsin L by IrThy, which exhibited low sensitivity to heparin, was associated with a decrease in inhibition of up to ~10%. 

The electrostatic character of the heparin interactions involved in the modulatory effect was examined by increasing the ionic strength in the inhibition assay (Figure 5B). The presence of 0.3 M NaCl either substantially or completely suppressed the heparin-induced changes observed in the inhibitory potency of IrThy. In addition, the control experiment with NaCl and without heparin suggests that electrostatic interactions also contributed to the interactions between IrThy and cathepsin K, as indicated by the ~20% reduction in inhibition.

In conclusion, GAGs modulate the inhibitory potency of IrThy against cysteine cathepsins in a complex manner. This modulatory effect is highly variable and depends on the type of target cathepsin, GAG, and GAG concentration employed. Most notably, IrThy inhibition of cathepsins K and V, both of which contain GAG-binding sites [32,33,34,35], was down- and upregulated by GAGs in opposite ways.

### 2.6. Spatial Structure of the IrThy C-Domain Determined by NMR Spectroscopy

To investigate the spatial structure of IrThy by NMR spectroscopy, we prepared full-length IrThy and its individual domains, IrThy-Nd and IrThy-Cd, in an *Escherichia coli* expression system for ^13^C and ^15^N isotope labeling. Correct protein folding was confirmed by comparing their 1D NMR spectra and inhibitory activity with recombinants produced in insect cells used in functional studies. 

For IrThy and IrThy-Nd, the poor quality of all standard 3D spectra indicated that the internal dynamics of both proteins contributed to a faster relaxation of the NH resonances, preventing the acquisition of reliable data. In contrast, the 3D spectra of IrThy-Cd were of sufficient quality to enable the assignment of ~85% of the backbone atoms and ~64% of all the atoms. Of the 68 IrThy-Cd residues, resonances for 17 residues were not detected. These included a stretch of Lys88−Val96, which is probably flexible and lacks contacts with more rigid structural elements. The other residues without backbone assignment were found in the proximity of cysteines Cys115 (Thr117), Cys139 (Ser136, Gly137, and Thr140), and Cys147 (Arg144, Asp145, His146, and Cys147 itself); this can be attributed to the dynamics of disulfide conformation on a millisecond timescale, resulting in a decrease in signal intensities. 

Interproton distance restraints were obtained from NOESY spectra (see Section 4). A dense network of inter-residue restraints was observed within the β-sheet-containing region Tyr109−Ala130; 18, 15, and 10 restraints were generated for residues Val123, Asp124, and Pro125, respectively. This points to a significant stabilization of this region, including the L2 loop (Figure 6B). As additional restraints, we also incorporated experimental mass spectrometry data on the pairing of cysteines in IrThy-Cd, namely Cys81–Cys103, Cys115–Cys139, and Cys122–Cys147, into the structure calculation protocol (Appendix A). The final refinement yielded 30 solution structures of IrThy-Cd selected on the basis of low overall energy and minimal constraint violations (Figure 6C). Atomic coordinates and experimental constraints were deposited in the Protein Data Bank (accession code: 8R6T) and the Biological Magnetic Resonance Data Bank (BMRB) (accession code: 34883).

A representative spatial structure of IrThy-Cd is presented in Figure 6A. The molecule adopts a wedge-shaped fold typical of thyropins and similar to those of the human p41 fragment (PDB code: 1ICF [6]) and bullfrog saxiphilin (PDB code: 6O0F [8]), the only thyropins whose structures are known. Figure 6B,D shows the structure-based sequence alignment and structural comparison of IrThy-Cd with p41 and the saxiphilin domains 1 and 2; the Cα RMSD values are within the range of 4.9−5.6 Å, and the sequence identities are within the range of 26−32%. While the architecture of IrThy-Cd is clearly similar to that of the other thyropins (Figure 6D), the high RMSD values can be attributed to the considerable flexibility of the loops and orientation of the structural regions. The fold is characterized by a conserved N-terminal α-helix (α1), an antiparallel β-sheet (β1–β2), and three loops (L1–L3), which are involved in the protease interaction (Figure 6A). In addition, IrThy-Cd incorporates another short C-terminal α-helix (α2). In terms of spatial arrangement, these segments constitute two subdomains: the first subdomain contains α1 and L1, and the second the rest of the thyropin molecule [6].

The disulfide bridges are critical to the overall fold of IrThy-Cd, considering the absence of a hydrophobic core, low content of secondary structures, and substantial conformational flexibility. Both IrThy-Cd subdomains are stabilized by the disulfide bridges Cys81–Cys103, Cys115–Cys139, and Cys122–Cys147, designated D1 through D3, respectively, in Figure 6A. Thus, IrThy-Cd conforms to the thyroglobulin subtype 1A domain, which has three disulfides [1]. In the first subdomain, the conserved D1 disulfide bridge near the N-terminus supports the formation of the L1 loop. The D2 and D3 bridges of IrThy-Cd, located in the second subdomain, connect the L2-containing segment with the L3-containing segment. Interestingly, the interloop disulfide bond D2 in IrThy-Cd is replaced by an intraloop disulfide in p41 and saxiphilin domain 1, where it forms a clamp within the L2 loop; this bridge is absent in saxiphilin domain 2. The higher number of interloop disulfides may provide greater rigidity for the second subdomain of IrThy-Cd. 

Loops L1, L2, and L3 form the reactive centers of the thyropins (Figure 6A). The tripartite wedge-shaped edge fills the active site cleft of the target proteases, as previously reported for the crystal structure of the p41 fragment complex with cathepsin L [6]. As the lowest-energy NMR structures of IrThy-Cd demonstrate, the L1- and L3-containing segments are the most flexible parts of the unbound thyropin molecule in solution (Figure 6C). The large L1 loop of IrThy-Cd features the conserved residues Gly101 and Pro105 (Figure 6B), which may be important for protease recognition specificity, as previously proposed for the L1 loops of thyropins [8]. The L2 loop, rather rigid in IrThy-Cd, is stabilized by the β1–β2 sheet and two disulfides. The β2 strand contains a part of the Trp–Cys–Val sequence motif conserved in the Tg1 domains (Figure 6B), which form the core of the second subdomain. The L3 loop is flanked by the D2 interloop disulfide followed by the C-terminal α2 helix; both of these features are unique to IrThy-Cd among the known thyropin structures.

## 3. Discussion

In this study, we provide a comprehensive structural and functional analysis of IrThy, an unusual member of the thyropin family of protease inhibitors that is expressed and secreted by the salivary glands of the hard tick *I. ricinus*. In general, the scientific understanding of thyropins in invertebrates is limited. To our knowledge, this is the first study to characterize parasite-derived thyropin at the molecular level. The IrThy molecule consists of two consecutive Tg1 domains, the N-domain and the C-domain. We identified an atypical disulfide pattern within the C-domain and also between both domains, forming an interdomain linkage. Intriguingly, the IrThy molecule lacks the complexity of structural modules common to many other multidomain Tg1-containing proteins. Our phylogenetic distribution analysis revealed that these notably short thyropin molecules (up to 180 residues), containing only two tandem Tg1 domains, are widespread in arthropods (including ticks) and entirely absent in mammals.

IrThy is a potent inhibitor of only three human cysteine cathepsins, namely V, K, and L, which belong to the cathepsin L subfamily of endopeptidases. IrThy does not inhibit other related cathepsins that possess diverse endo- and exopeptidase activities. Therefore, IrThy possesses a unique, narrow inhibitory specificity compared to other thyropins. In general, thyropins can be effective against endopeptidases, but also against exopeptidases. A wide range of cysteine cathepsins with endopeptidase activities, including cathepsins V, K, L, F, and S, are inhibited by the p41 fragment [5]. Inhibition of exopeptidases, such as cathepsins B and H, has been reported for the p41 fragment, equistatin, saxiphilin, and salmon egg thyropin [7,14,37,38].

The recombinant C-domain of IrThy (IrThy-Cd) exhibits the same inhibitory profile as full-length IrThy. We determined the spatial structure of IrThy-Cd by NMR spectroscopy, which allowed us to analyze the structural basis of the inhibitory specificity by comparing the IrThy-Cd structure with that of the p41 fragment complexed with cathepsin L [6]. The binding mode of the p41 fragment, which carries three inhibitory loops (L1, L2, and L3), enables important interactions with the S2–S1 subsite area as well as the surrounding loops located on the protease R-domain [6]. These interactions are understood to be negatively affected by two structural changes in IrThy-Cd, including the increased size of the L1 loop in the first subdomain and the shape of the second subdomain induced by the unique disulfide D2 (Figure 6B,D). In particular, these changes prevent binding to exopeptidases such as cathepsin H (an aminopeptidase) and cathepsin B (a carboxydipeptidase), whose partially obstructed active sites restrict access [39,40].

We observed that the recombinant N-domain (IrThy-Nd) displayed a broader selectivity pattern compared to the narrow inhibitory specificity of IrThy-Cd and the parental IrThy. In addition to cathepsins V, K and L, IrThy-Nd inhibited two other homologous proteases with endopeptidase activities, cathepsin F and papain. This suggests that the binding interactions of the N-domain are somewhat restricted by the closely positioned C-domain in the full-length IrThy scaffold. Nevertheless, our AlphaFold-derived model of IrThy (Appendix A) revealed that the two reactive centers of the N- and C-domains are oriented in opposite directions and can function simultaneously. A similar arrangement has been previously documented in the tandem Tg1 domains of saxiphilin, which bind two protease molecules with different inhibitory specificities [7]. 

Furthermore, we performed a preliminary cross-linking mass spectrometry (XL-MS) experiment with IrThy and cathepsin V to identify proximal structural regions in the inhibitor−protease complex (Appendix A). Our results indicated that IrThy preferentially binds to the active site of cathepsin V (added in an equimolar amount) through the reactive center on the N-domain, which exhibits a lower *K*_i_ value than the C-domain. However, further comparative XL-MS data collected under different concentration conditions and with other target cathepsins is required in order to better understand the IrThy binding process and the discrimination between domains. 

Our results show that several GAGs in their macromolecular forms modulated the inhibitory potency of IrThy in a complex manner. Widely distributed in host tissues, these sulfated polysaccharides regulate cysteine cathepsins, particularly their autocatalytic processing, pH stability, and interactions with specific macromolecular substrates [32,35,41,42,43,44,45]. In the presence of GAGs, the inhibitory potency of IrThy generally decreased against cathepsin K, increased against cathepsin V, and remained relatively unaffected against cathepsin L. As previously reported, GAGs participate in a variety of functional interactions with cathepsins K and V. For example, while the collagenolytic activity of cathepsin K requires interaction with chondroitin-4-sulfate, GAGs suppress the elastinolytic activities of both cathepsins K and V [32,35,46]. The molecular surfaces of cathepsins K and V contain positively charged patches for GAG binding [32,33,34,45]; however, these regions are less distributed in cathepsin L, leading to reduced interactions with GAGs [33]. We hypothesize that GAGs regulate IrThy inhibition by interacting with distinct GAG-binding sites on the target cathepsins, and that GAGs may also interact with positively charged regions found on the predicted surface model of IrThy, particularly on its N-domain (Appendix A). The arrangement of these intermolecular interactions can either facilitate or hinder IrThy binding.

Protease inhibitors from tick saliva injected into the host have been shown to modulate the host immune response and suppress blood clotting at the site of tick attachment and blood feeding [47]. To date, the only studied tick salivary inhibitors of cysteine cathepsins are members of the cystatin family, which exert immunosuppressive and anti-inflammatory effects on the host [23,24,25]. A recent comparative analysis of inhibitory specificity revealed that tick salivary cystatins differ from other family members in their high affinity for endopeptidases and limited effect on exopeptidases [24]. This biochemical feature makes tick salivary cystatins functionally similar to IrThy, exhibiting narrow inhibitory specificity against cysteine cathepsins with endopeptidase activities. It is therefore reasonable to assume that IrThy may play a role in immunomodulation and, based on the expression dynamics, that it acts in the initial phases of the tick–host interaction. In this context, cathepsins V, K, and L targeted by IrThy are known to be involved in the immune response and inflammation and are also present in the host skin [48,49,50,51,52,53]. Another clue to the potential physiological function of IrThy comes from a recent report on U24-ctenitoxin-Pn1a, a neurotoxin found in spider venom, that targets voltage-gated sodium channels (VGSCs) [54]. Interestingly, this protein contains two tandem Tg1 domains and is a close homolog of IrThy. The spider VGSC toxins are generally antinociceptive in an animal model [55], and it is tempting to speculate about the potential analgesic effect of IrThy at the tick bite site. Our future research will focus on identifying the exact physiological role of IrThy in tick–host interactions. 

## 4. Materials and Methods

### 4.1. Materials 

All protease substrates were purchased from Bachem (Bubendorf, Switzerland) with the exception of Abz-Phe-Arg-Phe(NO_2_)-OH from MP Biomedicals (Irvine, CA, USA). Human cathepsins L, K, and V were produced in the *Pichia pastoris* expression system as described previously [12,56,57]. Bovine cathepsins C and H were prepared as described in [58,59]; human cathepsin D was prepared as described in [60]. Bovine trypsin, chymotrypsin, and *Carica papaya* papain were purchased from Merck (Kenilworth, NJ, USA); human cathepsins F, B, and S were obtained from Enzo Life Sciences (New York, NY, USA); and human cathepsin X and legumain were obtained from R&D Systems (Minneapolis, MN, USA). GAGs were purchased from Sigma-Aldrich (St Louis, MO, USA), including heparin (H3393), chondroitin-4-sulfate (C9819), chondroitin-6-sulfate (C4384), dermatan sulfate (C3788), and dextran sulfate (D6924); heparin disaccharide (H1002) was purchased from Dextra (Reading, UK).

#### Ticks and Tick-Derived Materials

Adult *I. ricinus* ticks were collected in a forest near České Budějovice, Czech Republic. Ticks were kept at 24 °C and 95% humidity under a 15/9 h day/night regime. Adult *I. ricinus* females were fed on guinea pigs and either forcibly removed on the sixth day of feeding (half-fed) or allowed to fully feed (days 7 to 8). Saliva was collected from half-fed adult *I. ricinus* females as previously described [61]. Salivation was induced by applying pilocarpine solution to the tick scutum. Saliva was collected in 10-μL capillaries for 2 h in a wet chamber kept at 30 °C, pooled, and stored at −80 °C. The salivary glands were carefully removed without disrupting the epithelium and washed in phosphate-buffered saline (PBS). Tissue extracts (150 mg protein/mL) were prepared by homogenizing pooled tissues in 0.1 M sodium acetate pH 4.5, 1% CHAPS on ice. The extract was cleared by centrifugation (16,000× *g*, 10 min, 4 °C), filtered through Ultrafree MC 0.22 μm (Millipore, Bedford, MA, USA), and stored at −80 °C.

### 4.2. IrThy Cloning and Sequencing

Total RNA was isolated from the salivary glands of five adult *I. ricinus* females fed for six days, as previously described [62]. The RNA was reverse-transcribed into cDNA (0.5 μg RNA per 20 μL reaction) using the Transcriptor High-Fidelity cDNA Synthesis Kit (Roche Applied Science, Penzberg, Germany) with random hexamers and diluted 20 times in water. A 501 bp long fragment encoding IrThy was amplified from cDNA using specific forward 5′-atgctgaagtcaagtatagtagtg-3′ and reverse 5′-ctttcttgattagcagtggtcgc-3′ primers, which were designed based on a consensus sequence of thyropin homologs from *I. ricinus* sialome [26,63]. The sequence was deposited in GenBank under accession number PP107940. The resulting amplicon was isolated from the gel and cloned into the pCR4-TOPO vector (TOPO TA Cloning Kit, Thermo Fisher Scientific, Waltham, MA, USA). The isolated plasmid DNA was sequenced using the sequencing primers provided in the kit; the IrThy coding sequence resulted from the sequencing of seven individual clones. 

### 4.3. Production of Recombinant IrThy and its Domains in Insect Cells

S2 insect cells were used to produce full-length IrThy (residues 1–147) and its individual domains, IrThy-Nd (1–80) and IrThy-Cd (80–147). They were produced as fusion proteins containing an oligohistidine tag (6xHis) and a TEV protease cleavage site with an extension of Ser-Asn-Ala-Ala-Ser residues at the N-terminus. IrThy cDNA was PCR-amplified from the pCR4-TOPO vector containing the complete IrThy insert using the forward 5’-gctagcgtgccaactcgagtggcc-3´ and reverse 5’-gcggccgctaacagtggtcgcgc-3´ primers, which introduced the 5′ NheI and 3′ NotI restriction cloning sites (underlined). The PCR product was ligated into the pUC19 vector (Thermo Fisher Scientific) using SmaI restriction endonuclease. The N-glycosylation motif Asn-Leu-Thr106 was disrupted by Thr106 to Ala106 mutagenesis as described in [64]. pUC19 vectors containing individual single-domain inserts were prepared from full-length IrThy by inverse PCR [65] using the following primers: the forward 5′-tagcggccgcagggtaccgagctcg-3′ and reverse 5′-cttgcaggacttcagctgcctcgatgg-3′ primers for the C-domain deletion; the forward 5‘ aagtgtctggcagagcatcacgagaag-3′ and reverse 5′-gctagcggggggatcctctagagtcg-3′ primers for the N-domain deletion; all primers were 5′-phosphorylated. The resulting vectors were digested with NheI and NotI restriction endonucleases and ligated into the pMT/BiP/V5-His A plasmid (Thermo Fisher Scientific), which was modified to contain at the N-terminus a NheI restriction site and a sequence encoding the BiP signal sequence, His6 tag, and TEV cleavage site. 

Stably transfected insect cells were prepared as follows: Schneider 2 (S2) cells (Thermo Fisher Scientific) were cultured at 28 °C in Sf-900 II SFM medium (Thermo Fisher Scientific) containing 10% fetal bovine serum (Gibco, Thermo Fisher Scientific). Cells were co-transfected with the pMT/BiP/V5 vector containing IrThy-derived inserts (19 ng) and the pCoBlast selection vector containing the blasticidin resistance gene (1 ng) (Thermo Fisher Scientific) using the Calcium Phosphate Transfection Kit (Sigma Aldrich) according to the Drosophila Expression System protocol (Thermo Fisher Scientific). Stably transfected cells were selected with blasticidin (25 μg/mL) two weeks after transfection. S2 transfectants were expanded to a large volume (1.5 L) and cultured in Sf-900 II SFM medium to a density of 10^7^ cells/mL in a 6 L spinner flask, stirred at 120 rpm at 28 °C. Expression was induced with 1 mM CuSO_4_, and after 10 days, cells were removed by centrifugation (1000× *g*, 5 min, 10 °C). The supernatant was concentrated by lyophilization to a final volume of 150 mL and desalted on a Sephadex G25 column equilibrated with 10 mM Tris-HCl pH 8.0. 

Recombinant His-tagged proteins were purified by Ni^2+^ affinity chromatography on a HiTrap IMAC HP column (Cytiva, Marlborough, MA, USA) following the manufacturer’s protocol for elution with imidazole at pH 8.0. The His-tags were cleaved by TEV protease (enzyme:protein ratio 1:10, *w*/*w*) in 50 mM Tris-HCl pH 8.0, 0.5 mM EDTA, and 1 mM DTT for 12 h at 26 °C and removed by Ni^2+^ affinity chromatography on a HiTrap IMAC HP column (see above). This was followed by size-exclusion chromatography on a HiLoad 16/600 Superdex 75 column (Cytiva) equilibrated with 50 mM Tris-HCl, 300 mM NaCl, pH 8.0 (for IrThy and IrThy-Cd) or pH 7.0 (for IrThy-Nd). Protein purity was monitored by Laemmli SDS-PAGE on 15% polyacrylamide gels stained with Coomassie Brilliant Blue R-250. The purified proteins were buffer-exchanged into 20 mM Tris-HCl, 20 mM NaCl, pH 8.0 (for IrThy and IrThy-Cd) or pH 7.0 (for IrThy-Nd), concentrated using an Amicon Ultra-3K centrifugal unit (Millipore), and stored at −80 °C. These recombinants were used in all experiments except for the NMR structure determination.

### 4.4. Production of Recombinant IrThy and its Domains in E. coli 

The pUC19 vectors containing IrThy, IrThy-Nd, or IrThy-Cd inserts (see above) were digested with NcoI and NotI restriction endonucleases and ligated into the pETM-60 plasmid (EMBL, Heidelberg, Germany) in a frame with the NusA chaperone protein, His6-tag, and TEV protease cleavage site. Recombinant proteins were produced in SHuffle T7 Competent *E. coli* cells (New England BioLabs, Ipswich, MA, USA). Bacterial cultures were grown in ^15^N or ^13^C/^15^N isotope-containing optimized M9 minimal media (0.5 L) with 50 µg/mL kanamycin to an OD_600_ of 0.6. Expression was induced by the addition of isopropyl 1-thio-β-D-galactopyranoside to a final concentration of 0.4 mM. Cells were harvested by centrifugation (4000× *g*, 10 min, 4 °C) after 16 h of incubation at 16 °C with shaking at 200 rpm. Harvested *E. coli* cells were lysed using an EmulsiFlex-C3 homogenizer (Avestin, Ottawa, ON, Canada), and the lysate was clarified by centrifugation (12,000× *g*, 10 min, 4 °C). The recombinant His-tagged proteins were purified by Ni^2+^ affinity chromatography on a HiTrap IMAC HP column (Cytiva) using the manufacturer’s protocol for elution with imidazole at pH 8.0. The fusion tags were cleaved by TEV protease (see above) and removed by Ni^2+^ affinity chromatography on a HiTrap IMAC HP column (see above). The purified proteins were buffer-exchanged into 50 mM Tris-HCl pH 8.0, 150 mM NaCl, concentrated using an Amicon Ultra-3K centrifugal unit (Millipore), and stored at −80 °C. Protein purity was monitored by Laemmli SDS-PAGE (see above). 

### 4.5. Expression Analysis by Quantitative Real-Time PCR 

Total RNA was isolated from adult *I. ricinus* females during different stages of blood feeding and from different tissues of females fed for six days, as previously described [62]. The IrThy gene-specific qRT-PCR primers were designed using Primer3. The expression level of IrThy mRNA was measured by qRT-PCR using LightCycler 480 (Roche Applied Science) and SYBR green chemistry [62], with the forward 5′-atgagcaaatgcagtgcaac-3′ and reverse 5′-gcggttccactttctggat-3′ primers. Relative expression was normalized to *I. ricinus ferritin1* (AF068224) [66] according to the mathematical model of Pfaffl [67]. Data were obtained from three independent biological replicates (five females per replicate). Statistical significance was analyzed using the unpaired Student’s *t*-test and the one-way ANOVA test.

### 4.6. Mass Spectrometry Proteomic Analysis 

For the identification of native IrThy, proteins from the salivary gland extract and saliva of adult *I. ricinus* female were reduced, alkylated, and digested with trypsin. LC–MS/MS analysis of the digests was performed on an Ulti-Mate 3000 RSLCnano system (Dionex, Thermo Fisher Scientific) coupled to a Triple-TOF 5600 mass spectrometer equipped with a NanoSpray III source (ABSciex, Framingham, MA, USA). The peptides were separated on an Acclaim Pep-Map100 analytical column (3 μm, 150 × 0.75 mm; Thermo Fisher Scientific) by gradient elution in a 0.1% formic acid–acetonitrile mobile phase. Full mass spectrometry scans were recorded from 350 to 1250 *m/z*; up to 25 candidate ions per cycle were subjected to fragmentation; in MS/MS mode, fragmentation spectra were acquired from 100 to 1600 *m/z*. Mass data were processed using ProteinPilot 4.5 software (ABSciex).

For disulfide pairing analysis, recombinant IrThy, IrThy-Nd, or IrThy-Cd was digested with trypsin or Asp-N in the presence of cystamine; LC–MS/MS analysis of the digests was performed according to the protocol [68] described in Appendix A.

### 4.7. Phylogenetic Distribution Analysis

Phylogenetic distribution was analyzed for proteins with Tg1 domains (Pfam entry: PF00086) in the InterPro-hosted Pfam database [28]. Proteins containing two Tg1 domains and meeting the additional criteria specified in Section 2.2. were filtered using Python scripts.

### 4.8. Protease Inhibition Assays

Inhibition measurements were performed in triplicates in 96-well microplates (100 μL assay volume) at 37 °C. Recombinant inhibitors were preincubated with protease for 15 min, followed by the addition of a specific fluorogenic substrate (see below). The kinetics of product release were continuously monitored using an Infinite M1000 (Tecan, Männedorf, Switzerland) microplate reader at 365 nm excitation and 450 nm emission wavelengths (for AMC-containing substrates) or at 330 nm excitation and 410 nm emission wavelengths (for Abz-containing substrates). The progress curves of kinetic measurements were linear, indicating that the analysis was not affected by inhibitor degradation. IC_50_ values were determined from residual velocities using dose–response plots; nonlinear regression was fitted using GraFit software Version 7 (Erithacus, East Grinstead, UK), and the inhibition constants *K*_i_ were calculated using the Cheng–Prusoff equation [41,60]. 

Substrates and proteases were applied in assays as follows: Z-Phe-Arg-AMC in a concentration of 2.5 μM with 0.2 nM cathepsin L; 4.3 μM with 0.2 nM cathepsin K; 3.6 μM with 0.1 nM cathepsin V; 53 μM with 0.2 nM cathepsin B; 27 μM with 1.2 nM cathepsin F; 100 μM with 0.5 nM papain; 17 μM Z-Val-Val-Arg-AMC with 0.1 nM cathepsin S; 100 μM Arg-AMC with 1.9 nM cathepsin H; 87 μM Abz-Phe-Arg-Phe(NO_2_)-OH with 1.2 nM cathepsin X; 130 μM Gly-Arg-AMC with 0.4 nM cathepsin C; 70 μM Z-Ala-Ala-Asn-AMC with 1.2 nM legumain; 33 µM Abz-Lys-Pro-Ala-Glu-Phe-Phe(NO_2_)-Ala-Leu-NH_2_ with 3.5 nM cathepsin D; 67.8 μM Z-Phe-Arg-AMC with 0.6 nM trypsin; and 41.5 μM Suc-Ala-Ala-Phe-Arg-AMC with 0.02 nM chymotrypsin. The assay buffers were as follows: 50 mM sodium acetate pH 4.0 (for cathepsin D), pH 5.0 (for cathepsin X and legumain), or pH 5.5 (for cathepsins L, K, V, B, F, C, and papain); 50 mM MES pH 6.5 (for cathepsins S and H); and 50 mM Tris-HCl pH 8.0 (for trypsin and chymotrypsin). The buffers contained 0.1% polyethylene glycol 1500 and 2.5 mM dithiothreitol (for cysteine cathepsins and legumain), 50 mM NaCl (for cathepsin C), and 10 mM CaCl_2_ (for trypsin and chymotrypsin).

The inhibition mode was determined using an analogous activity assay with substrate concentrations of 1–10 μM for cathepsin K and 1–32 μM for cathepsin V; initial velocities of the product release were interpreted using a Lineweaver–Burk plot. The effect of pH on the inhibitory potency of IrThy was measured with cathepsins K and V (as described above) in 100 mM Britton–Robinson buffers within a pH range of 4.0–8.0. The effect of GAGs on the inhibitory potency of IrThy was measured with cathepsins K, L, and V (as described above) in the presence of various GAGs added to the preincubation mixture at a final concentration of 10 μg/mL. Heparin was also tested in a concentration range of 0.01–10 μg/mL. Where indicated, incubation was performed in the presence of 0.3 M NaCl.

### 4.9. Proteolytic Degradation of IrThy

Recombinant IrThy was incubated with cathepsins B, K, L, or V in 50 mM sodium acetate pH 5.5, 2.5 mM DTT, 0.1% polyethylene glycol 1500 at 37 °C. The reaction was stopped by the addition of 10 µM E-64 at different time points. A 100 μL aliquot contained 10 µg of IrThy (final concentration of 6 µM) and 2 µg or 0.5 ng of cathepsin (final concentrations of 0.8 μM or 0.2 nM, respectively). The reaction mixture was acetone-precipitated, separated by Laemmli SDS-PAGE (15% polyacrylamide gel), and stained with Coomassie Brilliant Blue R-250.

### 4.10. NMR Spectroscopy

NMR spectra were acquired at 25 °C using an 850 MHz Bruker Avance spectrometer equipped with a triple resonance (^15^N/^13^C/^1^H) cryoprobe. The protein sample (0.35 mL in a Shigemi tube) was measured in 50 mM deuterated Tris-HCl pH 8.0, 150 mM NaCl, and 5% D_2_O/95% H_2_O. A series of double- (Appendix A) and triple-resonance spectra [69,70] were recorded to obtain sequence-specific resonance assignment in NMRFAM-Sparky [71]. ^1^H–^1^H distance restraints were derived from spectra obtained by 3D ^15^N/^1^H NOESY-HSQC and ^13^C/^1^H NOESY-HMQC spectra using a NOE mixing time of 100 ms. The structure calculation was performed in CYANA [72] using NOESY data in combination with backbone torsion angle restraints generated from assigned chemical shifts using the TALOS+ program [73]. 

The combined automated NOE assignment and structure determination protocol (CANDID) was used for automatic NOE cross-peak assignment. Five cycles of simulated annealing combined with redundant dihedral angle restraints were then used to calculate a set of converged structures with no significant restraint violations (distance and van der Waals violations < 0.5 Å and dihedral angle constraint violations < 5°). The 30 structures with the least restraint violations were further analyzed using the Protein Structure Validation Software suite [www.nesg.org] (accessed on 11 December 2023). 

Statistics for the resulting structure are summarized in Appendix A. The structures, NMR restraints, and resonance assignments were deposited in the PDB (accession code: 8R6T) and BMRB (accession code: 34883) databases. Figures with structural representations were generated using PyMOL v2.3.2 (Schrödinger, New York, NY, USA) [74]. The structural comparison of IrThy-Cd with other thyropins provided the following Cα RMSD values (number of aligned Cα atoms is indicated): 5.36 Å (58 atoms) for the p41 fragment, 4.92 Å (50 atoms) for saxiphilin domain 1, and 5.62 Å (49 atoms) for saxiphilin domain 2.

## 5. Conclusions

Ticks are parasite vectors for a variety of viral and bacterial diseases in humans and domestic animals. Our study focuses on *Ixodes ricinus* ticks, vectors of Lyme borreliosis and tick-borne encephalitis. Tick saliva injected into the host modulates the physiological response at the tick bite site and facilitates the transmission of pathogens. Protease inhibitors from tick saliva are important effector proteins in this process and thus represent promising vaccination targets and pharmacological agents. Here, we characterize at the molecular level a new protease inhibitor, IrThy, found in *I. ricinus* saliva. It belongs to the thyropin family, which is poorly understood in parasites and pathogens. We provide a comprehensive analysis of IrThy, including its unique inhibitory specificity against only three human cysteine proteases, cathepsins V, K, and L, involved in immunity and inflammation, and the NMR structure that explains the functional properties. Based on the results obtained, we propose potential physiological roles for IrThy in host–parasite interaction at the tick bite site.

## Figures and Tables

**Figure 1 ijms-25-02240-f001:**
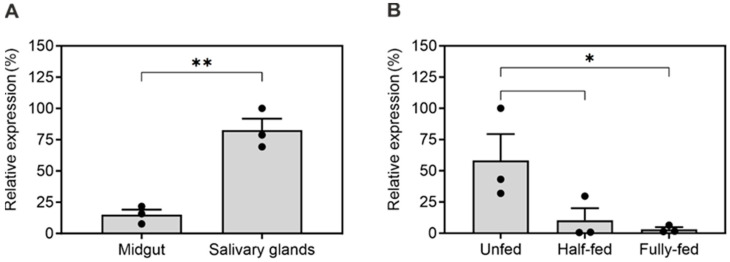
Transcriptional profiling of IrThy. Expression of IrThy was evaluated by qRT-PCR in adult female *Ixodes ricinus* ticks in (**A**) salivary glands and midgut from half-fed ticks and (**B**) whole-body homogenates at different stages of blood feeding. The mRNA transcript levels were normalized to the housekeeping gene *ferritin1*. Results represent the mean ± SD of biological triplicates (pooled samples), expressed relative to the highest measured value (100%); * *p* < 0.05, ** *p* < 0.005.

**Figure 2 ijms-25-02240-f002:**
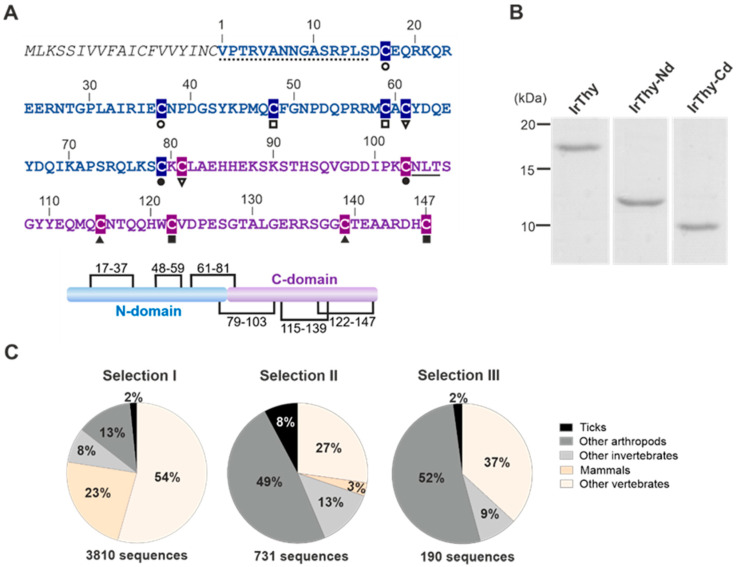
Sequence, evolution, and recombinant production of IrThy. (**A**) Amino acid sequence of IrThy featuring a signal peptide (italics), an unstructured N-terminal region (dotted), an N-terminal thyroglobulin-type 1 (Tg1) domain (blue), and a C-terminal Tg1 domain (purple). Residue numbering is according to the mature protein. The connectivity of cysteine residues (highlighted) is indicated by pairs of black symbols below the sequence. The predicted N-glycosylation site is underlined. The schematic diagram shows the organization of the domains (N- and C-domains) with six disulfide bridges (the black lines indicate cysteine residue connectivity). The signal peptide, the unstructured region, and the two Tg1 domains were predicted using SignalP 6.0, PrDOS, and InterPro, respectively. (**B**) Purified recombinant full-length IrThy and its individual N- and C-terminal domains (IrThy-Nd and IrThy-Cd), produced in the insect cell system, were resolved by SDS–PAGE and visualized by protein staining. (**C**) Phylogenetic distribution of proteins containing two Tg1 domains in invertebrates (including ticks and other arthropods) and vertebrates (including mammals). The sequences from three searches in the InterPro-hosted Pfam database are specified as follows: Selections I and II—the sequence length is not restricted, and other domain types may (Selection I) or may not (Selection II) be present in the molecules; Selection III—the sequence length is restricted to 180 residues, corresponding to proteins with only two consecutive Tg1 domains, such as IrThy.

**Figure 3 ijms-25-02240-f003:**
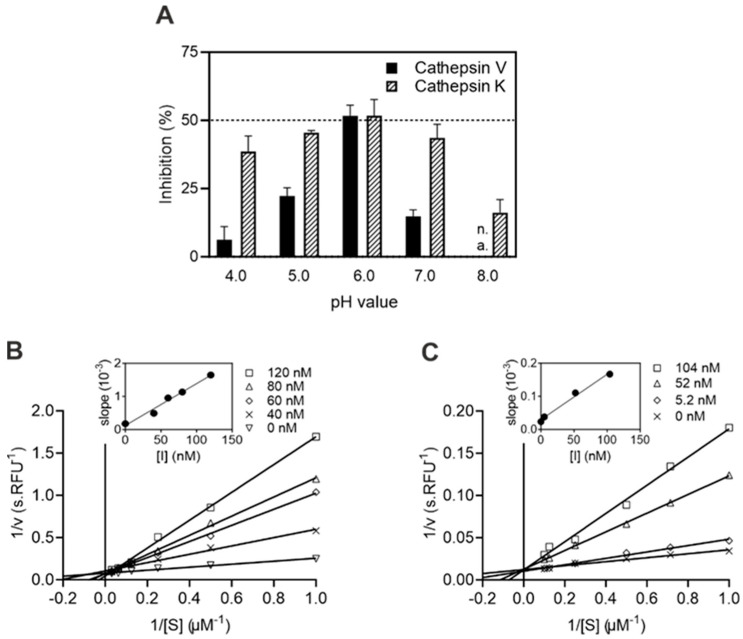
Characteristics of cysteine cathepsin inhibition by IrThy. (**A**) Effect of pH on the inhibitory potency of IrThy. The kinetic activity assay for cathepsins K and V with fluorogenic peptide substrates was performed at different pH values in the presence and absence of IrThy. The inhibitor was applied at a concentration providing ~50% inhibition at pH 6.0, and the % inhibition was calculated relative to an uninhibited control (0%) at the pH values indicated; means ± SD are given. Note that the pH value of 8.0 is outside the functional range of cathepsin V (n.a.). (**B**,**C**) Competitive mode of inhibition by IrThy. Lineweaver–Burk plots are presented together with secondary plots of the same data (inset) for (**B**) cathepsin V and (**C**) cathepsin K. The kinetic activity assay with fluorogenic peptide substrates was performed at pH 5.5. Means ± SD are given for triplicates.

**Figure 4 ijms-25-02240-f004:**
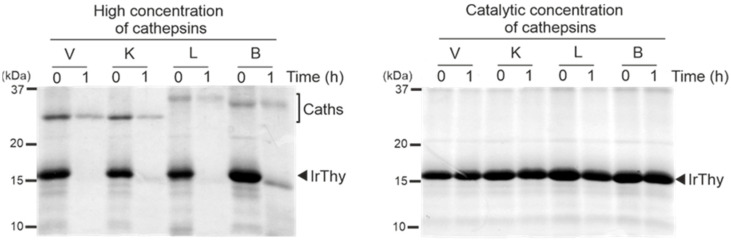
Proteolytic degradation of IrThy by target cathepsins V, K, and L, and a model non-target cathepsin B. IrThy was treated with high concentrations of cathepsins (an enzyme:inhibitor ratio of 1:5, *w*/*w*) or with catalytic concentrations of these cathepsins, corresponding to conditions in a kinetic inhibition assay (an enzyme:inhibitor ratio of 1:20,000, *w*/*w*). The reaction mixture was incubated at pH 5.5; the aliquots at time points 0 and 1 h were resolved by SDS–PAGE and visualized by protein staining. The positions of IrThy and cathepsins (Caths) are indicated.

**Figure 5 ijms-25-02240-f005:**
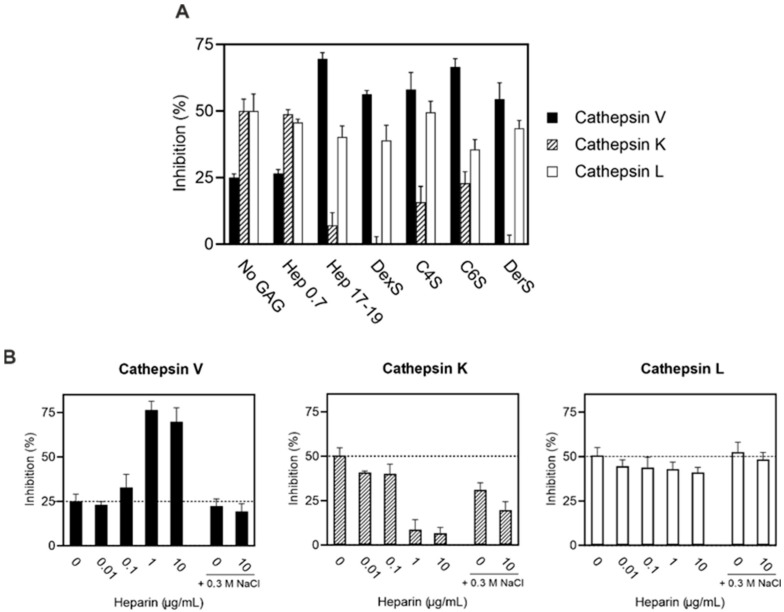
Effect of glycosaminoglycans (GAGs) on the inhibitory potency of IrThy. IrThy was applied at a concentration providing ~50% or ~25% inhibition in the absence of heparin. The kinetic activity assay with fluorogenic peptide substrates was performed at pH 5.5. The % inhibition was calculated relative to uninhibited controls (0%) with the same GAG concentrations (Appendix A); means ± SD are given. Comparative experiments without GAG are indicated as No GAG and 0 heparin. (**A**) The inhibitory potency of IrThy against human cathepsins V, K, and L in the presence of different GAGs, including heparin (Hep 17-19) and its disaccharide fragment (Hep 0.7), chondroitin-4-sulfate (C4S), chondroitin-6-sulfate (C6S), dermatan sulfate (DerS), and the GAG analog dextran sulfate (DexS). GAGs were applied at 10 µg/mL. (**B**) The inhibitory potency of IrThy against human cathepsins V, K, and L in the presence of various concentrations of heparin (0–10 µg/mL) and 0.3 M NaCl (where indicated).

**Figure 6 ijms-25-02240-f006:**
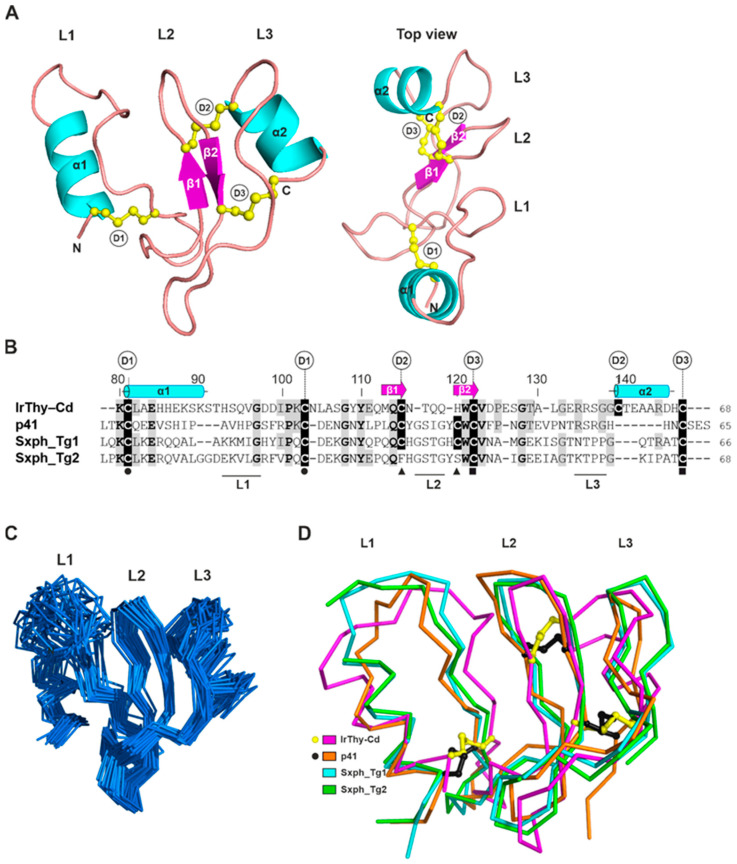
NMR solution structure of the IrThy C-domain (IrThy-Cd) and its comparison with other structurally characterized thyropins. (**A**) Representative three-dimensional structure of IrThy-Cd (PDB code: 8R6T) is depicted in a cartoon representation colored by secondary structural elements (α1–2 helixes, cyan; β1–2 strands, magenta). The N- and C-termini (N, C) and the three disulfide bridges (yellow sticks and balls) Cys81–Cys103 (D1), Cys115–Cys139 (D2), and Cys122–Cys147 (D3) are indicated. The L1, L2, and L3 loops are involved in the binding of the thyropin inhibitors to cysteine cathepsins [6]. (**B**) Structure-based sequence alignment of IrThy-Cd with structurally characterized thyropins, human p41 fragment (p41), and bullfrog saxiphilin domains (Sxph_Tg1 and Sxph_Tg2). Residues identical to those of IrThy-Cd are shaded in grey; fully conserved residues are in bold. Residue numbering is according to IrThy-Cd. Cysteine residues forming disulfide bridges are shaded in black and labeled D1–3 for IrThy-Cd; disulfide connectivity for the other thyropins is indicated by black circles, triangles, and squares below the alignment; note changes in the pairings compared to D2 of IrThy-Cd. The secondary structural elements of IrThy-Cd are depicted in cyan or magenta above the sequence and labeled the same as in (**A**). Three regions topped with the L1, L2, and L3 loops, which are responsible for the protease interaction, are indicated by horizontal black lines below the alignment based on the structure of the p41 complex with cathepsin L [6]. The alignment was generated using ClustalQ [36] and edited based on the structural superposition. (**C**) An ensemble of the 30 lowest-energy solution structures of IrThy-Cd in the ribbon representation is superimposed. The orientation of the molecule is the same as in (A) (left panel); the binding loops L1 to L3 are marked. (**D**) Overlay of IrThy-Cd (magenta) with structurally characterized thyropins in a ribbon representation, including the p41 fragment (orange, PDB code: 1ICF) and the saxiphilin domains 1 and 2 (cyan and green, respectively, PDB code: 6O0F). The positions of the disulfides are compared for IrThy-Cd (yellow sticks and balls) and p41 (black sticks and balls). The orientation of the molecules is the same as in (**A**) (left panel); the binding loops L1 to L3 are marked.

**Table 1 ijms-25-02240-t001:** Inhibitory specificity of IrThy and its individual domains. The inhibitory potency of full-length IrThy and its N- and C-terminal domains (IrThy-Nd and IrThy-Cd) was screened against mammalian host-derived cysteine cathepsins of the CA clan and archetypal representatives of protease clans. *K*_i_ values (mean ± SD) were determined by a kinetic activity assay using specific fluorogenic peptide substrates (see Section 4). The Merops database classification of the tested proteases (class/clan) and their mode of action are given. n.i.—no significant inhibition at 2 μM inhibitor concentration.

Enzymes	Enzyme Specificity, Protease Class/Clan		*K_i_* (nM)	
		IrThy	IrThy-Nd	IrThy-Cd
Host cathepsin proteases				
Cathepsin V	Endopeptidase, Cys/CA	34.7 ± 2.9	27.2 ± 3.1	40.7 ± 3.8
Cathepsin K	Endopeptidase, Cys/CA	56.9 ± 4.3	43.7 ± 5.4	53.1 ± 7.7
Cathepsin L	Endopeptidase, Cys/CA	201.4 ± 15.8	178.6 ± 6.7	795.1 ± 19.8
Cathepsin F	Endopeptidase, Cys/CA	n.i.	207.2 ± 14.2	n.i.
Cathepsin S	Endopeptidase, Cys/CA	n.i.	n.i.	n.i.
Cathepsin B	Endo- and carboxydipeptidase, Cys/CA	n.i.	n.i.	n.i.
Cathepsin X	Carboxypeptidase, Cys/CA	n.i.	n.i.	n.i.
Cathepsin C	Aminodipeptidase, Cys/CA	n.i.	n.i.	n.i.
Cathepsin H	Aminopeptidase, Cys/CA	n.i.	n.i.	n.i.
Model proteases				
Papain	Endopeptidase, Cys/CA	n.i.	153.4 ± 11.1	n.i.
Legumain	Endopeptidase, Cys/CD	n.i.	n.i.	n.i.
Cathepsin D	Endopeptidase, Asp/AA	n.i.	n.i.	n.i.
Chymotrypsin	Endopeptidase, Ser/PA	n.i.	n.i.	n.i.
Trypsin	Endopeptidase, Ser/PA	n.i.	n.i.	n.i.

## Data Availability

Atomic coordinates and experimental constraints were deposited in the Protein Data Bank (accession code: 8R6T) and the Biological Magnetic Resonance Data Bank (BMRB) (accession code: 34883).

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
