# Peer review of "An Unusual Two-Domain Thyropin from Tick Saliva: NMR Solution Structure and Highly Selective Inhibition of Cysteine Cathepsins Modulated by Glycosaminoglycans"

_ijms, 2024, doi:10.3390/ijms25042240_

Round 1

Reviewer 1 Report

Comments and Suggestions for Authors

Review of the Manuscript "An Unusual Two-Domain Thyropin from Tick Saliva: Highly Selective Inhibition of Cysteine Cathepsins Modulated by Glycosaminoglycans and an NMR Solution Structure"

The manuscript presents a comprehensive study on the characterization of a novel thyropin, IrThy, isolated from the saliva of Ixodes ricinus ticks. The authors investigate its structure, biochemical properties, inhibitory specificity against cysteine cathepsins, modulation by glycosaminoglycans (GAGs), and its potential role in tick-host interactions.

-The manuscript is well-organized and logically structured. The abstract provides a concise summary of the study, and the introduction clearly outlines the motivation and objectives. The results section is detailed and thorough, presenting experimental findings in a logical progression. However, some paragraphs are quite dense, and it might be beneficial to break them down for better readability.

-The investigation into the inhibitory specificity of IrThy against cysteine cathepsins is thorough and well-supported by experimental data. The modulation of IrThy's activity by GAGs is a particularly interesting aspect, and the data presented are compelling. However, the authors could elaborate more on the biological implications of GAG modulation in the context of tick-host interactions.

-The determination of the spatial structure of the IrThy C-domain using NMR spectroscopy is a significant achievement. The comparison with other thyropins adds valuable insights into the structural diversity within this family. However, a more detailed discussion on the functional implications of the structural features would enhance the manuscript.

-The language used is generally clear and concise. However, some sentences are complex, and simplifying the language in certain sections could improve accessibility for a broader readership.

In summary, the manuscript presents a well-executed study with valuable insights into a novel thyropin from tick saliva. Addressing the minor suggestions mentioned above would enhance the manuscript's clarity and impact. The research contributes significantly to the field of parasite-host interactions and may have broader implications for the development of therapeutic interventions.

Reviewer 2 Report

Comments and Suggestions for Authors

In this article, Matoušková and collaborators present an in-depth functional and structural analysis of Ir-thyropin (IrThy), an unusual thyropin that is secreted by Ixodes ricinus ticks in their saliva. The authors identified the protein in the salivary gland and saliva of the ticks using LC–MS/MS. By using qRT-PCR, they showed that the mRNA is predominantly expressed in the salivary glands of unfed female ticks and is secreted into the tick saliva, suggesting its involvement in the early stages of tick–host interaction. The authors used NMR to study the protein's structure and found that glycosaminoglycans (GAGs) change the protein's ability to inhibit cysteine proteases. I found very interesting the findings involving the modulation by GAGs, in particular the opposite effect observed for cathepsins K and V.

The article is very well written, and the results support the conclusions. In my opinion, the section Results is a little extensive and contains parts of the discussion. However, I understand the authors wanted to contextualize the results obtained. The article is attractive for readers and, in my opinion, represents an excellent contribution to the area. I have only a few minor comments, as follows:

Minor comments

In my opinion, the title could be paraphrased. Perhaps something like “An Unusual Two-Domain Thyropin from Tick Saliva: NMR Solution Structure and Highly Selective Inhibition of Cysteine Cathepsins Modulated by Glycosaminoglycans”

Wasn't the cDNA sequence obtained in this work deposited in GenBank? I think it is necessary to make the deposit, even if the sequence is similar to other deposited sequences.

In my opinion, the authors could include in the Supplementary Material the sequence of the IrThy cDNA and perhaps the alignments of the protein with sequences JAA67758.1 and JAP74612.1

The colors in Figure 2C are not completely adequate for daltonic people.

Figure’s 2 caption: “The signal peptide, the unstructured region, and the two Tg1 domains were predicted using SignalP 6.0, PrDOS, and InterPro, [respectively].”

Reviewer 3 Report

Comments and Suggestions for Authors

The manuscript by Matouskova and collaborators is a comprehensive study on the expression, localization, function and structure of a 2-domain thyrotropin from a tick, Ixodes ricinus, which is the host of Lyme's disease bacteria. The authors have used a plethora of techniques to characterize this protein, which is stabilized by disulfide bonds intra- and inter-domain.

Thyrotropin is released through the saliva at the site of the tick's bite and inhibits, by binding, at least three human cathepsins, namely V, K and L. The inhibition is affected differently (i.e. increased or relieved) by the presence of external GAGs. The enzymology experiments are sound, however there is a little incongruence in the buffer used to assess the inhibition of legumain and aminopeptidases. In fact the buffer contains 2.5 mM DTT, which is able to reduce all the S-S bonds, included those of Thyrotropin, therefore the non-inhibition of these proteases (reported in Table 1) might be due to the unfolding of IrThy. Did the Authors check by Circular Dichroism or NMR whether the structure of IrThy is maintained in those conditions or not ? A figure in the supplementary part should be added to support the data in Table 1.

The protein was recombinantly expressed in insect and in bacteria cells, however in both cases the putative N-glycosylation site had been mutated. Given that the activity of IrThy is affected by external GAG, the question arises whether an internal glycosylation would affect as well the binding and specificity to cathepsins V, K  and L.

Another minor concern refers to the conclusions: the Authors infer a role in host-pathogen interactions for IrThy. This is not what the study is about, the only fair conclusion is a role on a sort of facilitation of the blood meal at the site of the tick's bite, but none of the experiments points towards a role of IrThy on the % of release of the Lyme's bacteria.

Reviewer 4 Report

Comments and Suggestions for Authors

# 1

Lines 231-234 „No apparent degradation of IrThy was observed using these catalytic concentrations of cathepsins, with an enzyme:inhibitor ratio of 1:20,000 (w/w). On the contrary, incubation of IrThy with cathepsin concentrations increased by four orders of magnitude (1:5 ratio, w/w) led to the substantial degradation of IrThy by all three cathepsins.”

a) In the text of the paper (also in the description of methods) I could not find information on what were the concentrations of enzyme used in the study ? Only the ratio of enzyme to protein tested is given, not the absolute value of the concentration of at least one of the components tested.

(b) The images of the gels shown in Figure 4 do not show anything. To show that degradation is occurring or not, SDS-PAGE should be performed for a mixture of proteins at the start of incubation and after some time. The method used is not able to show the products of degradation, everything is based only on observation of the signal intensity of the substrate. If the authors had used such a broad spectrum of methods, especially HPLC and mass spectrometry, experiments identifying degradation products could have been performed (see reference 12).

c) It would also have been extremely interesting and valuable to investigate whether any of the enzymes tested by the authors that were not inhibited by the protein under study were substrates for the enzyme (see Table 1). 

Why did the authors use such a monstrous concentration difference (1:20000) for the study ? In the reference 12 tested there is a similar enzyme-inhibitor system, where degradation of the inhibitor molecule occurs. In this work (ref 12) authors also used a significant concentration difference, but it was a ratio of 1:550 (enzyme:inhibitor) at most. With such a huge disparity in concentrations, does the kinetic data obtained still make sense ?

# 2

Reference 12 presents a mathematical model to describe an enzymatic reaction in which two processes occur synchronously: substrate degradation and inhibitor degradation. Why did the authors, assuming inhibitor degradation, use the classical kinetic model ?

# 3

Figure 5: In the experiments related to GAGs, there is no control experiment to show that GAGs alone have no effect (or have) on the activity of the enzymes studied. The authors provide literature references for studying the interaction of enzyme inhibitors with GAGs, but there is no information on the GAGs enzyme interaction. However, it has long been established that GAGs affect the activity of cathepsins for example see:

Sage J., et al Binding of chondroitin 4-sulfate to cathepsin S regulates its enzymatic activity. Biochemistry2013;52(37):6487–6498. doi: 10.1021/bi400925g

Almeida P. C., at al Cathepsin B activity regulation. Heparin-like glycosaminoglycans protect human cathepsin B from alkaline pH-induced inactivation. Journal of Biological Chemistry2001;276(2):944–951. doi: 10.1074/jbc.M003820200

Control experiments results should be shown what effect GAGs have on enzyme activity, so that the effect of the proteins and GAGs tested can be reliably determined.

# 4

One wonders how the authors were able to interpret NMR spectra in solution containing Tris ? Did the authors use deuterated Tris ? I am referring to the interpretation of the proton spectra of NOESY, TOCSY, COSY. 

#5

Since the authors postulate that the protein has a domain structure and consists of two domains with probably very similar structures (see Figure 1S), it would be useful to add the sequence of the N-terminal domain to panel B of Figure 6 with a few sentences of commentary on the sequence differences/similarity of the two domains.

#6

The title of the paper is rather unfortunate. The current title suggests that the spatial structure determination method modulates the activity of the protein ?? Would the results of kinetic experiments depend on the fact that the structure of the domain would be determined by another method such as X-rar crystallography ? I guess the activity depends on the structure itself, not on the method by which this structure was determined.

In summary, I have no major comments on the structural part of the work. However, the part on biochemical studies requires additional experiments or analysis:

(a) a clear demonstration that the protein under study is degraded in the presence of enzymes that are simultaneously inhibited by the protein.

b) a discussion supported by literature data that the use of such high excesses of the inhibitor in relation to the enzyme does not distort the kinetic data obtained. Possibly conducting an experiment with the selected enzyme for several enzyme:inhibitor ratios and showing that the use of such huge excesses does not distort the description of the studied reaction.

c) Using equations that take into account the complexity of the system under study (enzyme+inhivitor/substrate+substrate).

d) Conducting experiments showing what effect GAGs have on the activity of the enzymes used in the work.

Comments on the Quality of English Language

None

Round 2

Reviewer 4 Report

Comments and Suggestions for Authors

I thank the authors for revisions to the manuscript. I have no further comments

Comments on the Quality of English Language

None